# A ring-like accretion structure in M87 connecting its black hole and jet

Ru-Sen Lu[1,2,3]✉, Keiichi Asada[4]✉, Thomas P. Krichbaum[3]✉, Jongho Park[4,5], Fumie Tazaki[6,7], Hung-Yi Pu[4,8,9], Masanori Nakamura[4,10], Andrei Lobanov[3], Kazuhiro Hada[7,11]✉, Kazunori Akiyama[12,13,14], Jae-Young Kim[3,5,15], Ivan Marti-Vidal[16,17], José L. Gómez[18], Tomohisa Kawashima[19], Feng Yuan[1,20,21], Eduardo Ros[3], Walter Alef[3], Silke Britzen[3], Michael Bremer[22], Avery E. Broderick[23,24,25], Akihiro Doi[26,27], Gabriele Giovannini[28,29], Marcello Giroletti[29], Paul T. P. Ho[4], Mareki Honma[7,11,30], David H. Hughes[31], Makoto Inoue[4], Wu Jiang[1], Motoki Kino[14,32], Shoko Koyama[4,33], Michael Lindqvist[34], Jun Liu[3], Alan P. Marscher[35], Satoki Matsushita[4], Hiroshi Nagai[11,14], Helge Rottmann[3], Tuomas Savolainen[3,36,37], Karl-Friedrich Schuster[22], Zhi-Qiang Shen[1,2], Pablo de Vicente[38], R. Craig Walker[39], Hai Yang[1,21], J. Anton Zensus[3], Juan Carlos Algaba[40], Alexander Allardi[41], Uwe Bach[3], Ryan Berthold[42], Dan Bintley[42], Do-Young Byun[5,43], Carolina Casadio[44,45], Shu-Hao Chang[4], Chih-Cheng Chang[46], Song-Chu Chang[46], Chung-Chen Chen[4], Ming-Tang Chen[47], Ryan Chilson[47], Tim C. Chuter[42], John Conway[34], Geoffrey B. Crew[13], Jessica T. Dempsey[42,48], Sven Dornbusch[3], Aaron Faber[49], Per Friberg[42], Javier González García[38], Miguel Gómez Garrido[38], Chih-Chiang Han[4], Kuo-Chang Han[46], Yutaka Hasegawa[50], Ruben Herrero-Illana[51], Yau-De Huang[4], Chih-Wei L. Huang[4], Violette Impellizzeri[52,53], Homin Jiang[4], Hao Jinchi[54], Taehyun Jung[5], Juha Kallunki[37], Petri Kirves[37], Kimihiro Kimura[55], Jun Yi Koay[4], Patrick M. Koch[4], Carsten Kramer[22], Alex Kraus[3], Derek Kubo[47], Cheng-Yu Kuo[56], Chao-Te Li[4], Lupin Chun-Che Lin[57], Ching-Tang Liu[4], Kuan-Yu Liu[4], Wen-Ping Lo[4,58], Li-Ming Lu[46], Nicholas MacDonald[3], Pierre Martin-Cocher[4], Hugo Messias[51,59], Zheng Meyer-Zhao[4,48], Anthony Minter[60], Dhanya G. Nair[61], Hiroaki Nishioka[4], Timothy J. Norton[62], George Nystrom[47], Hideo Ogawa[50], Peter Oshiro[47], Nimesh A. Patel[62], Ue-Li Pen[4], Yurii Pidopryhora[3,63], Nicolas Pradel[4], Philippe A. Raffin[47], Ramprasad Rao[62], Ignacio Ruiz[64], Salvador Sanchez[64], Paul Shaw[4], William Snow[47], T. K. Sridharan[53,62], Ranjani Srinivasan[4,62], Belén Tercero[38], Pablo Torne[64], Efthalia Traianou[3,18], Jan Wagner[3], Craig Walther[42], Ta-Shun Wei[4], Jun Yang[34] & Chen-Yu Yu[4]

The nearby radio galaxy M87 is a prime target for studying black hole accretion and jet formation[1,2]. Event Horizon Telescope observations of M87 in 2017, at a wavelength of 1.3 mm, revealed a ring-like structure, which was interpreted as gravitationally lensed emission around a central black hole[3]. Here we report images of M87 obtained in 2018, at a wavelength of 3.5 mm, showing that the compact radio core is spatially resolved. High-resolution imaging shows a ring-like structure of $8.4^{+0.5}_{-1.1}$ Schwarzschild radii in diameter, approximately 50% larger than that seen at 1.3 mm. The outer edge at 3.5 mm is also larger than that at 1.3 mm. This larger and thicker ring indicates a substantial contribution from the accretion flow with absorption effects, in addition to the gravitationally lensed ring-like emission. The images show that the edge-brightened jet connects to the accretion flow of the black hole. Close to the black hole, the emission profile of the jet-launching region is wider than the expected profile of a black-hole-driven jet, suggesting the possible presence of a wind associated with the accretion flow.

On 14–15 April 2018, we performed very-long-baseline interferometry (VLBI) observations of M87 with the Global Millimetre VLBI Array (GMVA) complemented by the phased Atacama Large Millimetre/submillimetre Array (ALMA) and the Greenland Telescope (GLT) at a wavelength of 3.5 mm (86 GHz; Supplementary Information section 1). The addition of the phased ALMA and GLT to the GMVA significantly improved the north–south resolution (by a factor of around 4) and baseline coverage in the direction perpendicular to the M87 jet. In Fig. 1, we show the resulting maps of M87, with a triple-ridged jet emerging from a spatially resolved radio core, which appears as a faint ring, with two regions of enhanced brightness in the northward and southward sections of the ring (Supplementary Information sections 2–4).

The most important feature of the image in Fig. 1a is the spatially resolved radio core. With the nominal resolution of our VLBI array, we

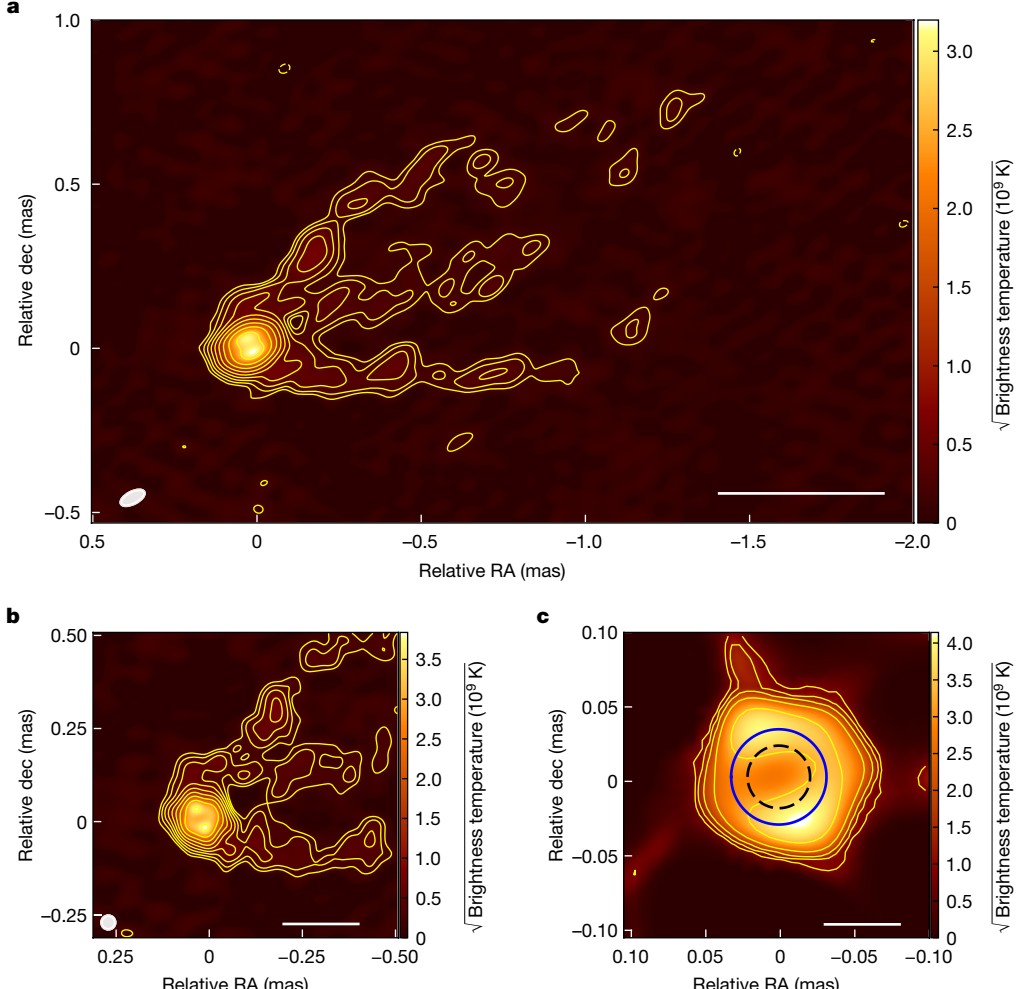

**Fig. 1 | High-resolution images of M87 at 3.5 mm obtained on 14–15 April 2018. a**, Uniformly weighted CLEAN (ref. [6]) image. The filled ellipse in the lower-left corner indicates the restoring beam, which is an elliptical Gaussian fitted to the main lobe of the synthesized beam (fullwidth at half-maximum = 79 μas × 37 μas; position angle = −63°). Contours show the source brightness in the standard radio convention of flux density per beam. The contour levels start at 0.5 mJy per beam and increase in steps of factors of 2. The peak flux density is 0.18 Jy per beam. **b**, The central region of the image as shown in **a**, but the image is now restored with a circular Gaussian beam of 37 μas size (fullwidth at half-maximum), corresponding to the minor axis of the elliptical beam in **a**. The peak flux density is 0.12 Jy per beam. The contour levels start at 0.4 mJy per beam and increase in steps of factors of 2. **c**, A magnification of the central core region using regularized maximum likelihood (RML) imaging methods. Contours start at 4% of the peak and increase in steps of factors of 2. The solid blue circle of diameter 64 μas denotes the measured size of the ring-like structure at 3.5 mm, which is approximately 50% larger than the EHT 1.3-mm ring with a diameter of 42 μas (dashed black circle)[4]. For each panel, the colour map denotes the brightness temperature $T$ in kelvin, which is related to the flux density $S$ in jansky as given in the equation $T = \lambda^2 (2k_B \Omega)^{-1} S$, where $\lambda$ is the wavelength, $k_B$ is the Boltzmann constant and $\Omega$ is the solid angle (shown on a square-root scale). The CLEAN images are the mean of the best-fitting images produced independently by team members, and the RML image is the mean of the optimal set of SMILI images (Supplementary Information section 3). dec, declination; RA, right ascension. Scale bars, 0.5 mas (**a**), 0.2 mas (**b**) and 50 μas (**c**).

see two bright regions of emission oriented in the north–south direction at the base of the northern and southern jet rails (Fig. 1a). Motivated by an obvious minimum (null) in the visibility amplitudes (Supplementary Figs. 10 and 11), we applied newly developed imaging methods that can achieve a higher angular resolution. This was done with and without subtracting the outer jet emission, to have a robust assessment of the parameters of the core structure (Supplementary Information section 3). From these images and by comparing ring- and non-ring-like model fits in the visibility domain, we conclude that the structure seen with the nominal resolution is the signature of an underlying ring-like structure with a diameter of $64^{+4}_{-8}$ μas (Supplementary Information sections 5–7), which is most apparent in slightly super-resolved images (Fig. 1b,c). Adopting a distance of $D = 16.8$ Mpc and a black hole mass of $M = 6.5 × 10^9 M_\odot$ (where $M_\odot$ is the solar mass)[4], this angular diameter translates to a diameter of $8.4^{+0.5}_{-1.1}$ Schwarzs-

child radii ($R_s = 2GM/c^2$, where $G$ is the gravitational constant, $M$ the black hole mass and $c$ the speed of light). On the basis of imaging analysis and detailed model fitting, we found that a thick ring (width ≳ 20 μas) is preferred over a thin ring (Supplementary Information). We note that the observed azimuthal asymmetry in the intensity distribution along the ring-like structure may (at least partly) be due to the effects from the non-uniform ($u$, $v$) coverage (Supplementary Information section 4), which also would explain the north–south dominance of the emission in the ring. Moreover, this double structure may also mark the two footpoints of the northern and southern ridge of the edge-brightened jet emission, which is seen further downstream. We note that previous GMVA observations[5]—without the inclusion of ALMA and the GLT—had a lower angular resolution, which was insufficient to show the ring–jet connection, but it is seen in the present images. We further note that the published 1.3-mm images did not

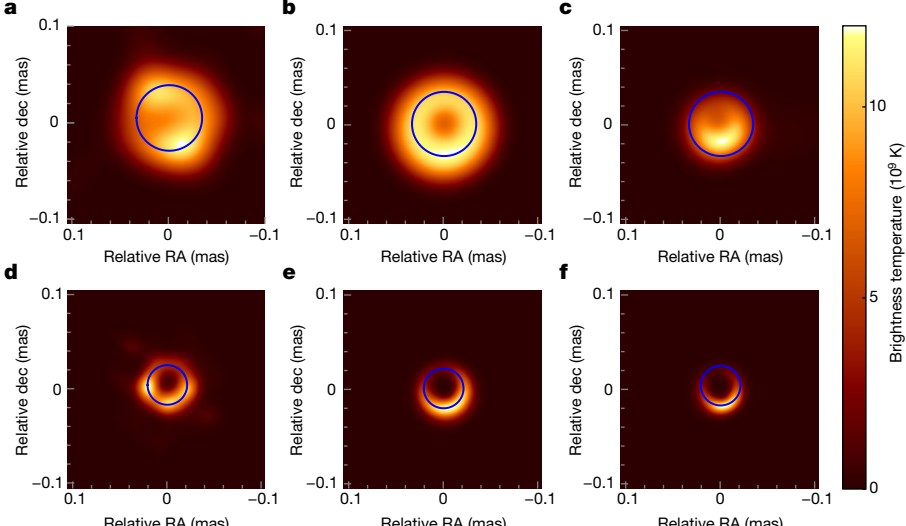

**Fig. 2 | RML images and model images at 3.5 mm and 1.3 mm. a–f**, RML images (**a**,**d**) and model images (**b**,**c**,**e**,**f**) obtained at 3.5 mm (**a–c**) and 1.3 mm (**d–f**). **a**, The 3.5-mm image obtained on 14–15 April 2018 is the same as in Fig. 1c but shown on a linear brightness scale. **b**,**e**, The thermal synchrotron model from the accretion flow assumes synchrotron emission from electrons with a Maxwellian energy distribution. **c**,**f**, The non-thermal synchrotron model from the jet region assumes synchrotron emission from electrons with a power-law energy distribution. **d**, The 1.3-mm EHT image obtained on 11 April 2017, reconstructed with the publicly available data[9] and imaging pipeline[6] using the EHT-imaging library[26]. Note that the differences in the azimuthal intensity

distribution in the two observed images are probably because of time variability and/or blending effects with the underlying jet footpoints. Although the morphology of both models is consistent with the observations at 1.3 mm (**e** and **f**), the larger and thicker ring-like structure at 3.5 mm can be understood by the opacity effect at longer wavelengths[27], preferentially explained by thermal synchrotron absorption from the accretion flow region (**b**). For comparison, reconstructed and simulated images are convolved with a circular Gaussian beam of 27 μas (3.5 mm) and 10 μas (1.3 mm) and are shown in a linear colour scale. The blue circle denotes the measured ring diameter of 64 μas at 3.5 mm and 42 μas at 1.3 mm.

reveal the inner jet emission because of $(u, v)$-coverage limitations[6] (see also recent re-analysis results[7,8]).

The ring-like structure observed at 3.5 mm differs from the one seen at 1.3 mm. The ring diameter at 3.5 mm ($64^{+4}_{-8}$ μas) is about 50% larger than that at 1.3 mm (42 ± 3 μas; ref. [4]). This larger size at 3.5 mm is not caused by observational effects (for example, calibration or $(u, v)$ coverage) and is already obvious from the $(u, v)$-distance plot of the visibilities (Supplementary Figs. 10 and 11). We note that the location of the visibility minimum, which scales inversely with the ring size, at 3.5 mm is at around 2.3 Gλ (Supplementary Information section 6). At 1.3 mm, the first visibility minimum is seen at a significantly larger $(u, v)$ distance of about 3.4 Gλ for the Event Horizon Telescope (EHT) data[9]. We find that the brightness temperature of the ring-like structure at 3.5 mm is approximately 1–2 × 10$^{10}$ K and the total compact flux density is roughly 0.5–0.6 Jy (Supplementary Table 2).

The reported fine-scale structure of the M87 jet base is substantially different from the classic morphology of radio-loud active galactic nuclei, characterized by a compact, unresolved component (core), from which a bright, collimated jet of plasma emanates and propagates downstream. Figure 1 shows a spatially resolved radio core with a ring-like structure and a triple-ridge jet structure[10] emerging to the west, with sharp gaps of emission between the ridges. Such a triple-ridge structure has been seen on larger scales (≳100$R_s$) in previous observations[5]. The location of the central ridge, which has an intensity of about 60% of that of the outer jet ridges, suggests the presence of a central spine, which emerges from the ring centre. The jet expands parabolically along a position angle of approximately −67° (Supplementary Information section 8), which is consistent with the jet morphology seen in previous studies[5]. Although previous images at 7 mm and 3.5 mm show some evidence for counterjet emission[5,11], we did not find any significant emission from a counterjet in this 2018 observation (upper limit of about 1 mJy per beam within 0.1–0.3 mas), possibly owing to its low brightness and limitations in the dynamical range.

Because we observed a ring-like structure, it is natural to assume that the black hole is located at its centre. Given the measured brightness temperature of about 10$^{10}$ K being typical for active galactic nuclei cores, synchrotron emission is believed to be responsible for the 3.5-mm ring-like structure. At 1.3 mm, it has been shown that the emission is always strongly lensed into the observed ring shape, regardless of whether it originates near the equatorial plane associated with the accretion flow or the funnel wall jet (jet sheath)[12]. As shown below, our observations at 3.5 mm can now constrain the spatial location and energy distribution of the electrons that are responsible for the millimetre emission.

The 2017 EHT observations have confirmed the nature of the accreting black hole in M87 to be in the low-Eddington regime, which is well described by a radiatively inefficient accretion flow (RIAF)[1,12]. On the basis of these studies, we model the spectral energy distribution and morphology of the horizon-scale structure assuming the emission is dominated either by the jet or by the accretion flow. This is done by applying a general relativistic radiative transfer to general relativistic magnetohydrodynamic simulations for an RIAF surrounding a rotating black hole (Supplementary Information section 9). The boundary between the accretion flow and jet is defined as the surface where the magnetic energy density equals the rest-mass energy density of the fluid (that is, $b^2/\rho c^2 = 1$; where $b$ is magnetic field strength, $\rho$ the plasma mass density and $c$ the speed of light). In the funnel region, where $b^2/\rho c^2 > 1$, synchrotron emission from electrons with a power-law energy distribution is assumed. Otherwise, where $b^2/\rho c^2 < 1$, synchrotron emission from electrons with a Maxwellian energy distribution is considered.

The properties of the non-thermal synchrotron model (from the jet) and the thermal synchrotron model (from the accretion flow) are normalized to fit the core flux density at 1.3 mm observed by the EHT[12]. For both models, the plasma around the black hole is optically thin at 1.3 mm. The resultant model images (Fig. 2e,f) are consistent with the observed morphology in terms of flux density, ring

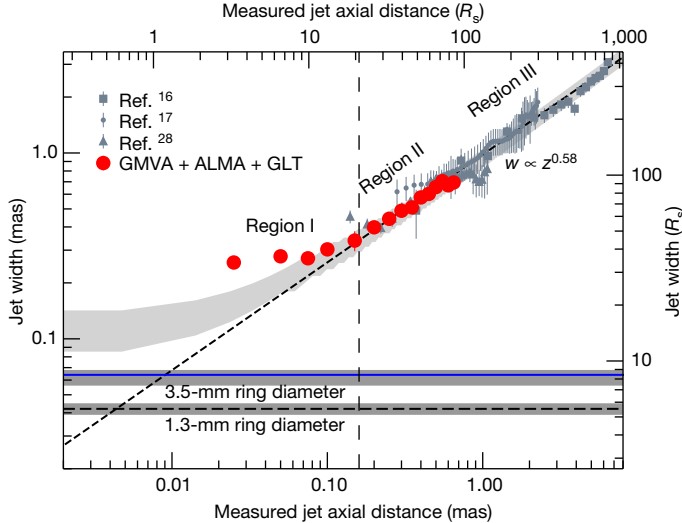

**Fig. 3 | Jet collimation profile.** Red filled circles mark the measured jet transverse width for the observations reported here. The error bars (1σ) are within the symbols (see Supplementary Information section 8 for more details on measuring the jet width). Grey filled squares, dots and triangles denote previous measurements of the width on larger scales[16,17,28], for which a power-law fit with a fixed power-law index of 0.58 is shown by the dashed line. The vertical dashed line marks the position at which the intrinsic half-opening angle $\theta$ of the fitted parabolic jet equals the jet viewing angle of $\theta_v = 17°$ (that is, boundary condition for a down-the-pipe jet[29]). The horizontal blue solid line marks the measured diameter of the ring at 3.5 mm, whereas the horizontal black dashed line marks the ring diameter measured with the EHT at 1.3 mm. In each case, the shaded area denotes the corresponding measurement uncertainty. The light-grey-shaded area denotes the outermost streamlines of the envelope of the parabolic jet from theoretical simulations (projected for $\theta_v = 17°$; ref. [30]) that are anchored at the event horizon[19] for a range of black hole spins (dimensionless spin parameters, $a = 0.0$–$0.9$). The lower and upper boundaries of this shaded area correspond to the highest ($a = 0.9$) and lowest ($a = 0.0$) spin, respectively. As the jet footpoint is anchored at the event horizon, some flattening of the jet width profile is expected near the black hole. This is further enhanced by geometrical projection effects in the region where the intrinsic jet half-opening angle ($\theta$) is larger than the jet viewing angle ($\theta_v$). The quasi-cylindrical shape in region I requires some change in the physical conditions to connect the innermost Blandford–Znajek jet from the event horizon to the upstream jet (region II).

diameter and width (Fig. 2d). In both models, the ring-like structure observed at 1.3 mm is dominated by lensed emission around the black hole.

At 3.5 mm, the plasma in both models becomes optically thick because of synchrotron self-absorption, resulting in a ring-like structure (Fig. 2b,c), diameter of which is larger than that at 1.3 mm. However, owing to the different emissivity and absorption coefficients for thermal and non-thermal synchrotron emission[13], the diameter of the resulting ring-like structure at 3.5 mm for the non-thermal model (Fig. 2c) would be smaller (≳30%) than our observed value. By contrast, the thermal model (Fig. 2b) is able to produce a ring-like structure consistent with the 3.5-mm observations (Fig. 2a), suggesting that the thermal synchrotron emission from the accretion flow region plays an important part in the interpretation of the 3.5 mm GMVA observations.

We note a marginal variability of the 1.3-mm flux density between April 2017 and April 2018 (ref. [14]). With the assumption that the overall ring size (determined by the black hole) observed at 1.3 mm in April 2017 did not change significantly[3,15], a comparison of the 1.3-mm and 3.5-mm images with the model predictions allows us to conclude that the larger ring size at 3.5 mm indicates the detection of an accretion flow, which is affected by synchrotron self-absorption (opacity) effects.

Our 2018 images allow us to study the jet collimation below the roughly 0.8 mas (about $100\,R_s$) scale in detail (Fig. 3). We note a change in the parabolic expansion near the ring (≲0.2 mas, region I), where the measured jet width forms a plateau and becomes larger than the parabolic jet profile seen further downstream (≳ 0.2 mas; regions II and III)[5,16,17].

The observed parabolic shape is consistent with a black-hole-driven jet through the Blandford–Znajek[18] process[19]. We note that the Blandford–Znajek jet model can produce a quasi-symmetric structure of limb-brightened jet emission if the black hole spin is moderately large ($a \gtrsim 0.5$), whereas the disk-driven jet model cannot[20]. Following previous studies[19], we examine the envelope of the Blandford–Znajek jet (light-grey-shaded area, Fig. 3). The observed jet width in the innermost region (region I in Fig. 3), however, is larger than this expected Blandford–Znajek jet envelope. We point out that a wide opening angle Blandford–Znajek jet launched from a strongly magnetized accretion flow (the so-called magnetically arrested disk)[21] may have difficulty in explaining this excess jet width. Therefore, such width-profile flattening suggests an extra emission component outside the Blandford–Znajek jet.

In addition to the jet, high-mass loaded, gravitationally unbound and non-relativistic winds have been found in RIAF simulations[22,23]. They are driven by the combination of centrifugal force[24] and gas and magnetic pressure[23] and are considered as an essential component collimating the Blandford–Znajek jet into a parabolic shape[19,25]. Non-thermal electrons accelerated by physical processes such as magnetic reconnection and shocks presumably exist in the wind. The synchrotron radiation of these non-thermal electrons may be responsible for this extra emission component[24] outside the Blandford–Znajek jet.

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

# Article

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

¹Shanghai Astronomical Observatory, Chinese Academy of Sciences, Shanghai, People's Republic of China. ²Key Laboratory of Radio Astronomy, Chinese Academy of Sciences, Nanjing, People's Republic of China. ³Max-Planck-Institut für Radioastronomie, Bonn, Germany. ⁴Institute of Astronomy and Astrophysics, Academia Sinica, Taipei, Taiwan, ROC. ⁵Korea Astronomy and Space Science Institute, Daejeon, Republic of Korea. ⁶Simulation Technology Development Department, Tokyo Electron Technology Solutions, Oshu, Japan. ⁷Mizusawa VLBI Observatory, National Astronomical Observatory of Japan, Oshu, Japan. ⁸Department of Physics, National Taiwan Normal University, Taipei, Taiwan, ROC. ⁹Center of Astronomy and Gravitation, National Taiwan Normal University, Taipei, Taiwan, ROC. ¹⁰Department of General Science and Education, National Institute of Technology, Hachinohe College, Hachinohe City, Japan. ¹¹Department of Astronomical Science, The Graduate University for Advanced Studies, SOKENDAI, Mitaka, Japan. ¹²Black Hole Initiative, Harvard University, Cambridge, MA, USA. ¹³Massachusetts Institute of Technology Haystack Observatory, Westford, MA, USA. ¹⁴National Astronomical Observatory of Japan, Mitaka, Japan. ¹⁵Department of Astronomy and Atmospheric Sciences, Kyungpook National University, Daegu, Republic of Korea. ¹⁶Departament d'Astronomia i Astrofísica, Universitat de València, Valencia, Spain. ¹⁷Observatori Astronòmic, Universitat de València, Valencia, Spain. ¹⁸Instituto de Astrofísica de Andalucía–CSIC, Granada, Spain. ¹⁹Institute for Cosmic Ray Research, The University of Tokyo, Chiba, Japan. ²⁰Key Laboratory for Research in Galaxies and Cosmology, Chinese Academy of Sciences, Shanghai, People's Republic of China. ²¹School of Astronomy and Space Sciences, University of Chinese Academy of Sciences, Beijing, People's Republic of China. ²²Institut de Radioastronomie Millimétrique, Saint Martin d'Hères, France. ²³Department of Physics and Astronomy, University of Waterloo, Waterloo, Ontario, Canada. ²⁴Waterloo Centre for Astrophysics, University of Waterloo, Waterloo, Ontario, Canada. ²⁵Perimeter Institute for Theoretical Physics, Waterloo, Ontario, Canada. ²⁶Institute of Space and Astronautical Science, Japan Aerospace Exploration Agency, Sagamihara, Japan. ²⁷Department of Space and Astronautical Science, The Graduate University for Advanced Studies, SOKENDAI, Sagamihara, Japan. ²⁸Dipartimento di Fisica e Astronomia, Università di Bologna, Bologna, Italy. ²⁹Istituto di Radio Astronomia, INAF, Bologna, Italy. ³⁰Department of Astronomy, Graduate School of Science, The University of Tokyo, Tokyo, Japan. ³¹Instituto Nacional de Astrofísica, Óptica y Electrónica, Puebla, Mexico. ³²Academic Support Center, Kogakuin University of Technology and Engineering, Hachioji, Japan. ³³Graduate School of Science and Technology, Niigata University, Niigata, Japan. ³⁴Department of Space, Earth and Environment, Chalmers University of Technology, Onsala Space Observatory, Onsala, Sweden. ³⁵Institute for Astrophysical Research, Boston University, Boston, MA, USA. ³⁶Department of Electronics and Nanoengineering, Aalto University, Aalto, Finland. ³⁷Metsähovi Radio Observatory, Aalto University, Kylmälä, Finland. ³⁸Observatorio de Yebes, IGN, Yebes, Spain. ³⁹National Radio Astronomy Observatory, Socorro, NM, USA. ⁴⁰Department of Physics, Faculty of Science, Universiti Malaya, Kuala Lumpur, Malaysia. ⁴¹University of Vermont, Burlington, VT, USA. ⁴²East Asian Observatory, Hilo, HI, USA. ⁴³University of Science and Technology, Daejeon, Republic of Korea. ⁴⁴Institute of Astrophysics, Foundation for Research and Technology, Heraklion, Greece. ⁴⁵Department of Physics, University of Crete, Heraklion, Greece. ⁴⁶System Development Center, National Chung-Shan Institute of Science and Technology, Taoyuan, Taiwan, ROC. ⁴⁷Institute of Astronomy and Astrophysics, Academia Sinica, Hilo, HI, USA. ⁴⁸ASTRON, Dwingeloo, The Netherlands. ⁴⁹Western University, London, Ontario, Canada. ⁵⁰Graduate School of Science, Osaka Metropolitan University, Osaka, Japan. ⁵¹European Southern Observatory, Santiago, Chile. ⁵²Leiden Observatory, University of Leiden, Leiden, The Netherlands. ⁵³National Radio Astronomy Observatory, Charlottesville, VA, USA. ⁵⁴Electronic Systems Research Division, National Chung-Shan Institute of Science and Technology, Taoyuan, Taiwan, ROC. ⁵⁵Japan Aerospace Exploration Agency, Tsukuba, Japan. ⁵⁶Department of Physics, National Sun Yat-Sen University, Kaohsiung City, Taiwan, ROC. ⁵⁷Department of Physics, National Cheng Kung University, Tainan, Taiwan, ROC. ⁵⁸Department of Physics, National Taiwan University, Taipei, Taiwan, ROC. ⁵⁹Joint ALMA Observatory, Santiago, Chile. ⁶⁰Green Bank Observatory, Green Bank, WV, USA. ⁶¹Astronomy Department, Universidad de Concepción, Concepción, Chile. ⁶²Center for Astrophysics | Harvard & Smithsonian, Cambridge, MA, USA. ⁶³Argelander-Institut für Astronomie, Universität Bonn, Bonn, Germany. ⁶⁴Institut de Radioastronomie Millimétrique, Granada, Spain. ✉e-mail: rslu@shao.ac.cn; asada@asiaa.sinica.edu.tw; tkrichbaum@mpifr-bonn.mpg.de; kazuhiro.hada@nao.ac.jp

## Data availability

The ALMA internal baseline data can be retrieved from the ALMA data portal (https://almascience.eso.org/alma-data) under the project code 2017.1.00842.V. The calibrated VLBI data used in this paper are used in a continuing project but can be made available on reasonable request from the corresponding authors.

## Code availability

Data processing and simulation softwares used in the paper, including AIPS (http://www.aips.nrao.edu/index.shtml), DIFMAP (https://sites.astro.caltech.edu/~tjp/citvlb), SMILI (https://github.com/astrosmili/smili) and the EHT-imaging library (https://github.com/achael/eht-imaging), are publicly available. The perceptually uniform colour maps for image visualization are available from the ehtplot library (https://github.com/liamedeiros/ehtplot). The general relativistic magnetohydrodynamic simulation and general relativistic radiative transfer are performed with publicly available codes using HARM (https://rainman.astro.illinois.edu/codelib) and ODYSSEY (https://github.com/hungyipu/Odyssey).

**Acknowledgements** R.-S.L. is supported by the Key Program of the National Natural Science Foundation of China (grant no. 11933007); the Key Research Program of Frontier Sciences, CAS (grant no. ZDBS-LY-SLH011); the Shanghai Pilot Program for Basic Research, Chinese Academy of Sciences, Shanghai Branch (JCYJ-SHFY-2022-013) and the Max Planck Partner Group of the MPG and the CAS. R.-S.L. thanks L. Blackburn, L. Chen, Y.-Z. Cui, L. Huang, R. S. de Souza and Y. Mizuno for discussions on data calibration and interpretation. J.P. acknowledges financial support through the EACOA Fellowship awarded by the East Asia Core Observatories Association, which consists of the Academia Sinica Institute of Astronomy and Astrophysics, the National Astronomical Observatory of Japan, Center for Astronomical Mega-Science, Chinese Academy of Sciences and the Korea Astronomy and Space Science Institute. H.-Y.P. acknowledges the support of the Ministry of Education Yushan Young Scholar Program, the Ministry of Science and Technology under grant no. 110-2112-M-003-007-MY2 and the Physics Division, National Center for Theoretical Sciences. K.H. is supported by JSPS KAKENHI grant nos. JP18H03721, JP19H01943, JP18KK0090, JP2101137, JP2104488 and JP22H00157. J.-Y. Kim acknowledges support from the National Research Foundation (NRF) of Korea (grant no. 2022R1C1C1005255). I.M.-V. acknowledges support from research project PID2019-108995GB-C22 of Ministerio de Ciencia e Innovacion (Spain), from the GenT Project CIDEGENT/2018/021 of Generalitat Valenciana (Spain), and from the Project European Union NextGenerationEU (PRTR-C17I1). F.Y. and H.Y. are supported by the Natural Science Foundation of China (grant nos. 12133008, 12192220 and 12192223) and China Manned Space Project (CMS-CSST-2021-B02). A.E.B. thanks the Delaney Family for their financial support through the Delaney Family John A. Wheeler Chair at Perimeter Institute. This work was supported in part by Perimeter Institute for Theoretical Physics. Research at Perimeter Institute is supported by the Government of Canada through the Department of Innovation, Science and Economic Development Canada and by the Province of Ontario through the Ministry of Economic Development, Job Creation and Trade. A.E.B. received further financial support from the Natural Sciences and Engineering Research Council of Canada through a Discovery Grant. S.K. acknowledges the Female Researchers Flowering Plan from MEXT of Japan, which supports research activities of female researchers. The research at Boston University was supported in part by NASA Fermi Guest Investigator grant no. 80NSSC20K1567. H. Nagai is supported by JSPS KAKENHI grant nos. JP18K03709 and JP21H01137. T.S. was partly supported by the Academy of Finland projects 274477, 284495, 312496 and 315721. P.d.V. and B.T. thank the support from the European Research Council through Synergy Grant ERC-2013-SyG, G.A. 610256 (NANOCOSMOS), and from the Spanish Ministerio de Ciencia e Innovación (MICIU) through project PID2019-107115GB-C21. B.T. also thanks the Spanish MICIU for funding support from grant nos. PID2019-106235GB-I00 and PID2019-105203GB-C21. This publication acknowledges project M2FINDERS, which is funded by the European Research Council (ERC) under the European Union's Horizon 2020 research and innovation programme (grant agreement no. 101018682). C.C. acknowledges support by the ERC under the Horizon ERC Grants 2021 programme under grant agreement no. 101040021. D.G.N. acknowledges funding from Conicyt through Fondecyt Postdoctorado (project code 3220195). This research has made use of data obtained using the Global Millimetre VLBI Array (GMVA), which consists of telescopes operated by the Max-Planck-Institut für Radioastronomie (MPIfR), IRAM, Onsala, Metsähovi Radio Observatory, Yebes, the Korean VLBI Network, the Greenland Telescope, the Green Bank Observatory (GBT) and the Very Long Baseline Array (VLBA). The VLBA and the GBT are facilities of the National Science Foundation (NSF) operated under cooperative agreement by Associated Universities. The data were correlated at the VLBI correlator of MPIfR in Bonn, Germany. This paper makes use of the following ALMA data: ADS/JAO.ALMA\#2017.1.00842.V. ALMA is a partnership of ESO (representing its member states), NSF (USA) and NINS (Japan), together with NRC (Canada), MOST and ASIAA (Taiwan) and KASI (Republic of Korea), in cooperation with the Republic of Chile. The Joint ALMA Observatory is operated by ESO, AUI/NRAO and NAOJ. The Greenland Telescope (GLT) is operated by the Academia Sinica Institute of Astronomy and Astrophysics (ASIAA) and the Smithsonian Astrophysical Observatory (SAO). The GLT is part of the ALMA–Taiwan project and is supported in part by the Academia Sinica (AS) and the Ministry of Science and Technology of Taiwan; 103-2119-M-001-010-MY2, 105-2112-M-001-025-MY3, 105-2119-M-001-042, 106-2112-M-001-011, 106-2119-M-001-013, 106-2119-M-001-027, 106-2923-M-001-005, 107-2119-M-001-017, 107-2119-M-001-020, 107-2119-M-001-041, 107-2119-M-110-005, 107-2923-M-001-009, 108-2112-M-001-048, 108-2112-M-001-051, 108-2923-M-001-002, 109-2112-M-001-025, 109-2124-M-001-005, 109-2923-M-001-001, 110-2112-M-003-007-MY2, 110-2112-M-001-033, 110-2124-M-001-007, 110-2923-M-001-001, and 110-2811-M-006-012. This research is based in part on observations obtained with the 100-m telescope of the MPIfR at Effelsberg, observations carried out at the IRM 30-m telescope by IRAM, which is supported by INSU/CNRS (France), MPG (Germany) and IGN (Spain), observations obtained with the Yebes 40-m radio telescope at the Yebes Observatory, which is operated by the Spanish Geographic Institute (IGN, Ministerio de Transportes, Movilidad y Agenda Urbana), and observations supported by the Green Bank Observatory, which is a main facility funded by the NSF operated by the Associated Universities. We acknowledge support from the Onsala Space Observatory national infrastructure for providing facilities and observational support. The Onsala Space Observatory receives funding from the Swedish Research Council through grant no. 2017-00648. This publication makes use of data obtained at the Metsähovi Radio Observatory, operated by Aalto University. It also makes use of VLBA data from the VLBA-BU Blazar Monitoring Program (BEAM-ME and VLBA-BU-BLAZAR; http://www.bu.edu/blazars/BEAM-ME.html), funded by NASA through the Fermi Guest Investigator Program.

**Author contributions** R.-S.L., K. Asada, T.P.K. and K.H. initiated the project and coordinated the research. P.H., M.I., M.-T.C., S.M., K. Asada, P.M.K., Y.-D.H., C.-C.H., D.K., P.A.R., T.J.N., N.A.P. and other engineers and technicians in East Asia made the GLT available for our observations. K. Asada, T.P.K., R.-S.L., I.M.-V., J.P. and E.R. worked on the VLBI scheduling and coordination of the observations, analysis and calibration of the data. K. Akiyama, K. Asada, K.H., J.-Y. Kim, T.P.K., R.-S.L., J.P., F.T. and A.L. worked on image reconstruction and model fitting. K. Asada, J.L.G., K.H., J.-Y. Kim, T.P.K., T.K., A.L., R.-S.L., M.N., J.P., H.-Y.P. and F.Y. worked on the scientific interpretation of the results. All authors approved the paper and contributed to writing the observing proposal, carried out the observations, produced and/or applied software tools for analysis and interpretation, and contributed to the general interpretation of the data.

**Competing interests** The authors declare no competing interests.

**Additional information**
**Correspondence and requests for materials** should be addressed to Ru-Sen Lu, Keiichi Asada, Thomas P. Krichbaum or Kazuhiro Hada.
