## [Peer review File · Nature]

Manuscript Title: A ring-like accretion structure in M87 connecting its black hole and jet

Reviewer Comments & Author Rebuttals

Reviewer Reports on the Initial Version:

Referees' comments:

Referee #1 (Remarks to the Author):

I have read the manuscript and supplementary material for this paper and believe that this article warrants publication in Nature after some very minor points discussed below are addressed by the authors. The work is timely and represents an important step forward in understanding the formation and collimation of AGN jets. I find the manuscript to be very well-written, the data-analysis methods clearly described, and the conclusions of immediate interest to the astronomical community.

The paper is based on a single-epoch GMVA+phased ALMA+GLT VLBI observation of the nearby radio galaxy M87 (aka 3C 274) at 3.5 mm (86 GHz) obtained in April 2018. This source has been imaged many times before at mm wavelengths, but the recent EHT imaging of the black hole shadow of this nearby, radio-bright galaxy at 1.3 mm make this source one of particular interest and importance. The new VLBI observation discussed in the paper benefits from the inclusion of both phased ALMA and the Greenland telescope in the VLBI experiment, adding long baselines and additional (u,v) coverage. The images constructed from these data employ a newly-developed reconstruction technique which is less sensitive to incomplete (u,v) coverage, and high resolution images are produced using SMILI in addition to images based on the standard VLBI package CLEAN. These improvements lead to a high-resolution image of the inner compact jet structure showing a ring-like emission feature not resolved in previous lower-resolution images. The authors proceed to attribute the site of this emission to the spatial domain linking the accretion disk surrounding the Black Hole and the portion of the downstream jet routinely imaged at lower frequencies, probing a region of the jet which has not been well-studied before. As part of the analysis, they identify the properties of the ring-like structure (diameter and width), and they further constrain the physical site of the emission region and the energy distribution of the electrons producing the millimeter emission by comparison with model simulations. Based on the wider-than-expected shape of the observed emission profile, produced by combining non-coeval data at several wavelengths, they propose that the excess emission near the BH compared to theoretical expectations could be indicative of a wind from the accretion flow. The paper carefully illustrates how the new data presented here can be effectively used within the framework of the VLBI baseline coverage currently available, and the authors present a convincing case for the scenario proposed.

The ABSTRACT clearly states the main results described in the paper: 1) the robust identification of a ring of emission similar to the feature recently identified by the EHT at 1.3 mm; and 2) construction and analysis of the emission profile at spatial distances inward of 100 Schwarzschild radii. The bulk of the paper is devoted to the identification of the properties of the ring-like structure in the 3 mm

data. The method for obtaining and reducing the data and tests of the validity of the various results are very clearly described and carefully presented to the reader in the supplementary material. This supplementary material is lengthy, but it is very useful to have this in-depth discussion included as part of the paper so that the reader can properly understand how the conclusions were reached.

While the paper is already of high quality, and the authors have anticipated and answered most questions which a reader might have, I suggest below a few minor text additions, revisions, and questions for consideration by the authors.

Figure 1a assumes an elliptical restoring beam while Figure 1b assumes a circular restoring beam which results in the blown-up image in Figure 1c discussed in the analysis (see also Supplement line 471). Will the authors please clarify for the reader what determines the choice of the restoring beam as this affects the geometry of the ring. As a more minor point, in the figure the dashed grey circle marking the corresponding ring in the EHT data is rather faint and would be more readily visible by using a darker shade of grey.

Figure 2 compares the new image at 3.5 mm (obtained 14-15 April 2018) with the published EHT image at 1.3 mm obtained a year earlier. These images show structure in the emission, and the paper in the supplementary material addresses how structure due to hot spots and the deduced ring feature are related. Visual comparison of the emission features alone identifies that these hot spots are not the same in the two emission maps. Both spatial differences in the emission due to the difference in the physical location of the emitting region as well as time-dependent structural changes in the emission at each of the two wavelengths may play some role in the patterns. Clearly high-resolution, coeval observations at the two wavelengths would be helpful. The authors have properly used the data they have in hand, but mention of these caveats might be a useful addition to the text.

To validate the preference for the adopted ring geometry in the discussion, Figure S11 in the supplementary material shows comparison of the emission in the q - V plane with model fits assuming ring emission from a variety of geometrical shapes. The caption for Figure S10 states that the values longward of $q=1.5$ are more sensitive to the inner compact structure. In this domain the thin and the thick ring model results only appear to differ in the jet-subtracted figure at $q>2.5$ where the data are sparse. Does restricting the test for the best value to this region provide too few data points for a robust test? In distinguishing and ruling out the models considered, I find the designation "thin ring" a bit misleading. What does "thin" mean here? Isn't this a simulation where the ring thickness is assumed to be negligible? Can the authors find a better label to designate this scenario? Also, do they have an explanation for why the data distribution is so flat in the q - V plane longward of $q=2.5$ in the jet-subtracted plot (Figure S11)? Also, all readers of the paper will not be familiar with VLBI terminology, so the authors need to define q in the captions for these two figures.

Figure 3 shows the jet collimation profile based on the measured jet width which includes the new data indicated by red filled circles. Regarding presentation, there are no error bars on the red points, or indication of the generic error associated with these measured values in the figure caption. Is this error within the symbol size? This information should be added to the figure or stated in the figure caption. Also, I presume that "observed distance" labeled on the abscissa means "measured", or are

these deprojected values? Additionally, the caption should include references for the top 3 symbols in the upper left portion of the figure. Further, I would have expected to see a comparison to results in Kim et al. 2018 (cited elsewhere but not in this specific context). Is there some reason why it was not included? Regarding interpretation, the deviations of these points from the simulation are an important new result, but the suggested explanation of a wind to account for a broader-than expected profile in the spatial region inward of 10 Schwarzschild radii is only marginally explored. Is stacking expected to fill in the emission and widen the emission profile as more high-resolution mm data are obtained in future experiments? If so, this will increase the discrepancy between data and simulation identified here and enhance the need for a plausible physical explanation.

Line 193 says that an outcome of the work is to constrain the energy distribution of the electrons responsible for the millimeter emission. The SED shown in the supplementary material (Figure S15) pertains to the electron distribution and would be better placed in the main body of the paper (as Figure 3 before the collimation profile.)

In Figure S2c some other color scheme should be adopted for this plot. It is difficult to distinguish the blue and the black points, especially where the density of points is relatively high.

Were the data for the calibrator 3C 279 used in the analysis? If so what was their role?

In Figure S15 do the authors have any suggestions for why the data labeled “historical” are so different from the more recent data which are very consistent?

While the paper has been unusually well-prepared, I noted a few places in the text where terms are not defined for the reader, or the idea is so briefly stated that it is obscure. These are listed below.

line 150: Please clarify what is meant by “two railed jet emission”?

line 191-192: what is the “funnel wall jet”.

lines 222- 225: Higher concentration would affect intensity. How does it affect size?

line 444: The comment that the ringlike structure can be explained by opacity effects is obscure and needs further explanation.

supplement line 131: “Possibilities” is an odd choice of word. Would “configurations” be better?

supplement line 142: “Basis” could be replaced with “mode”.

The text in supplement lines 213 and 430 both contain the word “opacity”. Presumably the authors mean opacity in the sky in the first usage, and opacity in the jet outflow in the second. It would be helpful to the reader for the authors to clarify what they have in mind.

Referee #2 (Remarks to the Author):

This paper presents new GMVA + ALMA + GLT observations at 3.5mm of the core of M87 and compares and contrasts the results with the 2017 EHT images at 1.3mm. The work is original, topical, well-presented, and of general interest, and so deserving of consideration for publication in Nature. I append below a list of recommendations, suggestions, and corrections (many of which are relatively minor) which I believe will help improve the paper.

My main concern is that the conclusions of the paper are, by necessity, based on a comparison of the 2017 EHT imaging with the GMVA+ALMA+GLT observations made one year later, under the assumption that the accretion disk has not changed significantly in the meantime. It is clear that the region around the core does change – the paper notes this is the most sensitive 3.5mm observation made, yet there is no sign of the counter-jet reported at earlier epochs. And we know that new jet components are on occasions ejected from the core. Accretion towards the black hole is unlikely to be steady, and the accretion disk could plausibly swell up and shrink back down from time to time, and the core flux density change as a result. The authors could mount a case, based on multi-wavelength monitoring, that M87 appears to have been in a similar state in 2018 and 2017, or perhaps present simulations that suggest the accretion disk size does not change dramatically under a variety of circumstances. I would need to see some consideration of this point before recommending the paper be accepted for publication.

I note that this paper, like the EHT imaging of M87, presents images that are averaged over an ensemble of plausible images produced independently by “several” members of the team. Perhaps it is not possible to be more precise about exactly how many independent analyses were conducted, but “several” is quite vague. Nine authors are mentioned in the “Author Contributions” as having worked on the image reconstruction and model fitting: perhaps that is sufficient, but I would encourage more quantitative statements where that is possible.

Line 108:

“In 2017, Event Horizon Telescope observations of M87 at the wavelength of 1.3 mm revealed a ring-like structure which was interpreted as gravitational lensed emission around a central black hole.”

I think this would read better moving “In 2017” into the sentence, e.g., “Event Horizon Telescope observations of M87 in 2017, at a wavelength of 1.3 mm, revealed a ring-like structure which was interpreted as gravitational lensed emission around a central black hole.”

Line 113:

“High resolution imaging reveals a ring-like structure of $8.4_{(-1.1)}^{(+0.5)}$ Schwarzschild radii in diameter, ~50% larger than that seen at 1.3 mm. The outer edge at 3.5 mm is larger than that at 1.3 mm.”

This would read better as

“High resolution imaging reveals a ring-like structure $8.4_{(-1.1)}^{(+0.5)}$ Schwarzschild radii in diameter, ~50% larger than that seen at 1.3 mm, with the outer edge of the ring at 3.5 mm also being larger than that at 1.3 mm.”

Line 117:

“The new images show that the edge–brightened jet connects to the accretion flow and the black hole.”

I advocate either “...connects to the accretion flow around the black hole.” or “...connects to the accretion flow.” The authors postulate in line 149 that the double structure may be the footpoints of the jet which (line 241) are anchored at the event horizon, but also provide good evidence that the effects of limited (u,v) coverage are important, so I would recommend not claiming the images show the jet connecting to the black hole (assuming that the event horizon can be considered the black hole itself...).

Line 124: “GMVA project code ML005”

This information is given in the Supplementary Methods, which is the better place for it, so can be omitted here.

Line 126:

“In Fig. 1, we display the resulting maps of M87, revealing a triple-railed jet emerging from a spatially resolved radio core which appears as a faint ring...”

I suggest “triple-ridged” rather than “triple-railed”

Line 136:

“This was done with and without subtracting the outer jet emission (Supplementary Information, Section 3).”

That is quite reasonable, but it would be appropriate to add a few words of explanation as to why it was considered necessary to do this.

Line 138:

“...we arrive at the conclusion that the structure seen with the nominal resolution, indeed is the signature of an underlying ring-like structure with a diameter of $64_{(-8)}^{(+4)} \mu\text{s}$ (see Supplementary Information for details), which becomes best visible in slightly super-resolved images.”

I suggest “is most apparent” rather than “becomes best visible”

Line 143:

“ $R_s=2G-M/c^2$ ”

Remove the dash between G and M

Line 149:

“On the other hand, this double structure may also mark the two 'footpoints' of the northern and southern ridge of the two railed jet emission, which is seen further downstream.”

It is previously noted that the jet is triple-ridged, so some consideration of why the central ridge does not have an associated footpoint may be appropriate, within this paragraph or the paragraph beginning on Line 163. It would be useful to point out here that the EHT imaging shows no jet emission at all, due to the limited region they considered in their imaging. It should also be noted whether (and why) the EHT image in Figure 2d was reconstructed adopting the same constraints as the EHT team (and stating what they are).

Line 156:

“This larger size at 3.5 mm is not caused by observational effects...”

It would be appropriate to state, or give examples of, the potential “observational effects” considered here.

Lines 157 and 160:

“...uv-distance...”

Would be better as (u,v)-distance

Line 169:

“The intensity of the central ridge is ~60% of that of the outer jet ridges, suggesting the presence of a central spine, which points towards the ring centre.”

At face value, the sentence seems to imply it is the lower intensity of the central ridge that suggests it is a central spine – if that is indeed the intention further explanation should be given. The location of the central ridge is sufficient to assume it is the jet spine, in which case “The location of the central ridge, which has an intensity ~60% of that of the outer jet ridges, suggests the presence of a central spine...” would suffice.

But does the central spine “point towards the ring centre” or away from it? Perhaps “which emerges from the ring centre” or “which is aligned with the ring centre” would be better, although in fact extrapolation of the innermost portion of the central ridge does not appear to pass through the ring centre. While “spine and sheath” models for jets have been considered for some years, to the best of my knowledge M87 is the only source which appears to display both edge-brightened sheath and interior spine. (And this result is arguably more robust, as it does not entail comparison with the EHT observation a year earlier.) The GMVA+ALMA+GLT image is much more convincing than those in the cited refs 5 & 7. It is probably beyond the scope of this paper to consider in more detail, but I trust a paper exploring the implications of this jet structure is planned.

Line 175:

“In order to understand the physical nature of the 3.5 mm ring-like structure, we first need to estimate the position of the black hole. It is well-known that the location of the unresolved core in VLBI images of AGN jets shows a frequency dependence (the so-called “core shift”, e.g., ref.10). For a synchrotron self-absorbed homogeneous jet the VLBI-core shifts upstream when observing at higher frequencies. Previous core shift measurements of M8711 also showed this dependence. From these measurements, the position of the jet apex was extrapolated to be located at a projected distance of $41 \pm 12 \mu\text{as}$ or $5.4 \pm 1.6 R_s$ (adopting the black hole mass and distance from EHT results⁴) upstream of the 7 mm (43GHz) radio core. Extrapolating the frequency dependent core position further would place the black hole at a projected distance of $\lesssim 3 R_s$ ($\sim 21 \mu\text{as}$) upstream of the VLBI core at 3.5 mm (where we neglect a small position offset between jet apex and the BH position). It is therefore natural to assume that the centre of the ring-like structure which we have observed at 3.5 mm is close to the actual black hole position.”

I think all bar the last sentence of this paragraph could probably be omitted. The “core shift” effect is relevant for the classic morphology described in Lines 163—166 but, as noted, the 3.5mm image considered here does not have an unresolved core. As we see a ring, it is natural to assume the black hole lies at its centre. The rest of the paragraph could be moved to the Supplementary Methods, or possibly omitted entirely.

Line 187:

“Given the typical brightness temperature of $\sim 10^{10}$ K, synchrotron emission is believed to be responsible for the 3.5 mm ring-like structure.”

As the measured brightness temperature is stated on line 161, this sentence would be better reworded as something like

“Given the measured brightness temperature of $\sim 10^{10}$ K is typical for that of AGN cores, synchrotron emission is believed to be responsible for the 3.5 mm ring-like structure.”

Line 194:

“Recent EHT observations have confirmed the nature of the accreting black hole in M87 to be in the low-Eddington regime, ...”

I suggest “The 2017 EHT observations...” rather than “Recent EHT observations...”

Line 200”

“The boundary between the accretion flow and jet is defined as the surface where the magnetic energy density equals to the fluid's rest-mass energy density...”

Either “...density equals the...” or “...density is equal to the ...”

Line 221:

“From the above discussion, the thermally-distributed electron model explains the larger ring diameter at 3.5 mm, under the constraints of the 1.3 mm and 3.5 mm flux density.”

At a minimum this should be “... 1.3 mm and 3.5 mm flux densities”, but better would be

“... under the constraints of the (2017 April) 1.3 mm and (2018 April) 3.5 mm flux densities.”

This would lead naturally to some consideration of whether, and how much, the core of M87 may have evolved in the intervening period, and the effect of such variability on the conclusions drawn.

Line 246:

“We point out that a laterally expanding wide opening angle BZ jet launched from a MAD accretion flow...”

Need to spell out MAD at its first usage.

Line 319:

“Data availability

The ALMA raw visibility data can be retrieved from the ALMA data portal under the project code 2017.1.00842.V. The reduced data that support the results of this study are available from the corresponding authors upon reasonable request.”

Some additional detail would be useful – is it the ALMA phased array data that is available? Are the ALMA internal baselines also available? Perhaps “The reduced data...” could be expanded to “The reduced GMVA + ALMA + GLT data...”, assuming that is correct.

Figure 1:

Line 422: Expand “p.a.”

Line 431: “For each panel, the colour map denotes the brightness temperature T in Kelvin, which is related to the flux density S in Jy ...”

When spelled out in full, a physical unit named after a person is not capitalised (even though its abbreviated form is!). So it should be "... brightness temperature T in kelvin ..." (and similarly could be "... the flux density S in jansky")

It would be appropriate to note in the caption that these images are the averages of the best fitting images produced independently by several team members, referring to the Supplementary Methods for more details.

Figure 3:

The inset or legend to the figure would be better with "ref 16" or "Asada and Nakamura 2016" rather than "AN16", and similarly for H13 and H16.

Line 458: "The light-grey shaded area denotes the outermost streamlines of the parabolic jet's envelope from theoretical simulations ... that are anchored at the event horizon for a range of black hole spins (dimensionless spin parameters: $a = 0.0 - 0.9$)."

It would be of interest to note which extreme of the shaded grey area corresponds to which extreme of the spin parameter.

Supplementary Methods

=====

Line 109:

"M87 was observed by the Global Millimetre VLBI Array (GMVA) in concert with the phased Atacama Large Millimetre/submillimetre Array (ALMA) and the Greenland Telescope (GLT)¹ at 3.5 mm (86 GHz; GMVA project code ML005). The GMVA is composed of the eight Very Long Baseline Array (VLBA) antennas equipped with 3.5 mm receivers, ..."

It would be appropriate to add how many of ALMA's dishes were phased up for these observations (only 37 of the 54 12-m dishes were phased up for the 2017 EHT observations), and also which array configuration ALMA was in at the time. It would also be useful to explicitly state which two VLBA antennas do not have 3.5mm receivers.

Line 153:

"We removed the instrumental polarisation effects from the GLT by applying, in the frame of the Radio Interferometer Measurement Equation..."

"...in the frame-work..." would be better

Line 190:

"We then derived and applied bandpass corrections using calibrator scans and performed the a priori amplitude calibration in the standard manner, using measured system temperatures and gain curves and including opacity corrections for each station."

Consider adding a brief description of what measurements the opacity corrections are based on.

Line 203:

"and closure phases of $\sim 180^\circ$ on ALMA-US-EU and ALMA-US-GLT triangles."

The use of US and EU is probably clear, but perhaps spell out in full for clarity.

Line 225:

“This was done independently by several authors.”

This is an important point. Is there any reason the number of independent authors (or teams) could not be given?

Line 238:

“are in good agreement with near in time VLBI observations of the Korea VLBI Network (KVN) at 3.5 mm...”

should be “Korean VLBI Network”

Line 276:

“We utilised weighted-L1 (wL1), total squared variation (TSV), and total variation (TV) regularizers for sparse imaging (see ref.9 for their mathematical definitions).”

It would be useful to include here a little more detail, for example

“...total squared variation (TSV, which favours smooth edges)...”

Line 298:

“For the other hyper parameters, we tried multiple values.”

A little more detail on how these multiple values were chosen would be useful.

Line 301:

“Since the chi-square values of all images now were reasonably low, a chi-square cutoff was not applied.”

should be “chi-squared”

Line 303:

“The resulting optimal set of images (216 images in total)...”

It would be of interest to note how many images in total were generated (and therefore what fraction were, and were not, considered optimal)

Line 309:

“We mainly used this source because the GLT did not observe 3C279.”

Presumably as 3C279 is too far south to observe with the GLT – but if that’s the case, then explicitly saying so would be better.

Line 324:

“With our high-resolution image, we also found that there is emission upstream of the brightness peak. Full details of these results will be published elsewhere.”

This is interesting, but with the saturated colour scale in Figure S5b it is not readily apparent. In any case it is peripheral to the main focus of this paper, so these two sentences could possibly be dropped.

Line 346:

“We postulate that this difference originates from: (i) the limited (u,v)-coverage of our data, and (ii) that the RML technique is better suited for super-resolution imaging than CLEAN.”

This would read better as “...(ii) the fact that the RML...”

Line 354:

“One is based on the real (u,v)-coverage, 10-second averaged data and the other is generated by adding artificial baselines based on two new stations (VLBA MK and KVN Yonsei).”

VLBA MK is not really new – it participated in the observation but no fringes were found for M87. Perhaps “two additional stations”? Or “adding artificial baselines to VLBA MK and KVN Yonsei.”

Line 381:

“Using this tentative position as the origin, we unwrapped the image azimuthally and obtained radial profiles for each azimuthal angle between 0 and 360°.”

Does this mean 360 radial profiles in one-degree steps?

Line 389:

“There are some radial profiles where the brightness monotonically decreases with radius (see, e.g., Fig. S6a).”

This is not clear to me – I do not see any radial profiles that decrease monotonically with radius from the centre of the ring in Fig. S6a!

Line 412:

“This fitting was applied to both the full self-calibrated dataset and to the jet-subtracted dataset. In each case, the data were averaged to 420 seconds.”

It would be appropriate to add a justification for this choice of averaging interval.

Line 451:

“In addition, we note the formal size of the inner depression as described by r_{in} is about 1.7 times larger than the inner radius of $\sim 13 \mu\text{as}$ measured from the EHT data²¹.”

I suggest “best-fit size” (or “formal best-fit size”)

Line 454

Altogether, the analysis presented above suggests the presence of a ring-like structure in the centre of M87 which has a mean radius of $\sim 67 \mu\text{as}$ and a width of $\sim 22 \mu\text{as}$.

“radius” should be “diameter”

Line 486:

“We performed an axisymmetric 2D GRMHD numerical simulation for magnetised plasma around a rotating black hole (dimensionless black hole spin parameter =0.9) by using the public code HARM”

Please add a justification for this choice of spin parameter

Line 499:

“For modelling the core emission and image structure of M87, we consider a black hole of 6.5 billion solar masses ...”

I recommend scientific notation, 6.5×10^9

Figure S7:

“Simulated (u,v)-coverage for the 2018 observations of M87 (blue) and for the artificially added

baselines based on VLBA MK in Hawaii, USA and KVN Yonsei in South Korea (red).”

Perhaps

“Simulated (u,v)-coverage for the 2018 observations of M87 (blue) and for the additional baselines to VLBA MK in Hawaii, USA and KVN Yonsei in South Korea (red), added for the purpose of investigating the effects of (u,v) coverage on the results.”

Figures S9 and S14.

The insets for both figures should use “With jet” and “Without jet” rather than “W jet” and “W/O jet”

References:

Reference 6: The volume number (which should appear in bold) is 285, and the page number (which should not be in bold) is 109.

Reference 14: The article id. of this paper is 95, not 5

There appears to be some inconsistency in both reference lists: in some cases both the first and last page numbers of articles are given, in other cases only the first page number is given (and of course in some cases only the article id. is required)

Author Rebuttals to Initial Comments:

Referees' comments:

Referee #1 (Remarks to the Author):

I have read the manuscript and supplementary material for this paper and believe that this article warrants publication in Nature after some very minor points discussed below are addressed by the authors. The work is timely and represents an important step forward in understanding the formation and collimation of AGN jets. I find the manuscript to be very well-written, the data-analysis methods clearly described, and the conclusions of immediate interest to the astronomical community.

The paper is based on a single-epoch GMVA+phased ALMA+GLT VLBI observation of the nearby radio galaxy M87 (aka 3C 274) at 3.5 mm (86 GHz) obtained in April 2018. This source has been imaged many times before at mm wavelengths, but the recent EHT imaging of the black hole shadow of this nearby, radio-bright galaxy at 1.3 mm make this source one of particular interest and importance. The new VLBI observation discussed in the paper benefits from the inclusion of both phased ALMA and the Greenland telescope in the VLBI experiment, adding long baselines and additional (u,v) coverage. The images constructed from these data employ a newly-developed reconstruction technique which is less sensitive to incomplete (u,v) coverage, and high resolution images are produced using SMILI in addition to images based on the standard VLBI package CLEAN. These improvements lead to a high-resolution image of the inner compact jet structure showing a ring-like emission feature not resolved in previous lower-resolution images. The authors proceed to attribute the site of this emission to the spatial domain linking the accretion disk surrounding the Black Hole and the portion of the downstream jet routinely imaged at lower frequencies, probing a region of the jet which has not been well-studied before. As part of the analysis, they identify the properties of the ring-like structure (diameter and width), and they further constrain the physical site of the emission region and the energy distribution of the electrons producing the millimeter emission by comparison with model simulations. Based on the wider-than-expected shape of the observed emission profile, produced by combining non-coeval data at several wavelengths, they propose that the excess emission near the BH compared to theoretical expectations could be indicative of a wind from the accretion flow. The paper carefully illustrates how the new data presented here can be effectively used within the framework of the VLBI baseline coverage currently available, and the authors present a convincing case for the scenario proposed.

The ABSTRACT clearly states the main results described in the paper: 1) the robust identification of a ring of emission similar to the feature recently identified by the EHT at 1.3 mm; and 2) construction and analysis of the emission profile at spatial distances inward of 100 Schwarzschild radii. The bulk of the paper is devoted to the identification of the properties of the ring-like structure in the 3 mm data. The method for obtaining and reducing the data and tests of the validity of the various results are very clearly described and carefully presented to the reader in the supplementary material. This supplementary material is lengthy, but it is very useful to have this in-depth discussion included as part of the paper so that the reader can properly understand how the conclusions were reached.

While the paper is already of high quality, and the authors have anticipated and answered most questions which a reader might have, I suggest below a few minor text additions, revisions, and questions for consideration by the authors.

We would like to sincerely thank this referee for carefully reading the paper and the positive overall review. We are very pleased that the reviewer found that the paper is already of high quality as is.

Figure 1a assumes an elliptical restoring beam while Figure 1b assumes a circular restoring beam which results in the blown-up image in Figure 1c discussed in the analysis (see also Supplement line 471). Will the authors please clarify for the reader what determines the choice of the restoring beam as this affects the geometry of the ring. As a more minor point, in the figure the dashed grey circle marking the corresponding ring in the EHT data is rather faint and would be more readily visible by using a darker shade of grey.

We thank the referee for this comment. We have clarified the choice of the restoring beam both in the caption of Figure 1 and in supplementary section 8 (after the original Supplement line 471). We agree with the referee that the dashed grey circle is rather faint. We have changed the colors for both circles and hope they are now better visible.

Figure 2 compares the new image at 3.5 mm (obtained 14-15 April 2018) with the published EHT image at 1.3 mm obtained a year earlier. These images show structure in the emission, and the paper in the supplementary material addresses how structure due to hot spots and the deduced ring feature are related. Visual comparison of the emission features alone identifies that these hot spots are not the same in the two emission maps. Both spatial differences in the emission due to the difference in the physical location of the emitting region as well as time-dependent structural changes in the emission at each of the two wavelengths may play some role in the patterns. Clearly high-resolution, coeval observations at the two wavelengths would be helpful. The authors have properly used the data they have in hand, but mention of these caveats might be a useful addition to the text.

This is a very good point and we agree with the referee that coeval observations at the two wavelengths would be helpful. We have put a caveat on this in the caption of Figure 2.

To validate the preference for the adopted ring geometry in the discussion, Figure S11 in the supplementary material shows comparison of the emission in the q - V plane with model fits assuming ring emission from a variety of geometrical shapes. The caption for Figure S10 states that the values longward of $q=1.5$ are more sensitive to the inner compact structure. In this domain the thin and the thick ring model results only appear to differ in the jet-subtracted figure at $q>2.5$ where the data

are sparse. Does restricting the test for the best value to this region provide too few data points for a robust test? In distinguishing and ruling out the models considered, I find the designation “thin ring” a bit misleading. What does “thin” mean here? Isn’t this a simulation where the ring thickness is assumed to be negligible? Can the authors find a better label to designate this scenario? Also, do they have an explanation for why the data distribution is so flat in the q - V plane longward of $q=2.5$ in the jet-subtracted plot (Figure S11)? Also, all readers of the paper will not be familiar with VLBI terminology, so the authors need to define q in the captions for these two figures.

We thank the referee for the comments. Please note that in Figure S10, the models are fitted to the data that include the contribution of the extended jet. Even in this case, the ring models are preferred by the data (Supplementary Table S1). Guided by this fitting, it can be seen that the jet contribution only becomes important at $q \lesssim 1.5$. The data in Figure S11 is more appropriate for discriminating the fitted models.

We note that the data beyond $q > 2.5$ are the best calibrated ones on ALMA baselines and the preference on the thick ring model is robustness as indicated by the chi-squared fitting statistics presented in Supplementary Table S2.

The referee is correct regarding the “thin ring”. We have introduced this “infinitesimally thin ring” at the beginning of supplementary section 6 and now refer to it as “thin ring” thereafter. We thank the referee for the suggestion on defining q . We have defined q when we first introduced it in the text (supplementary section 6) and mentioned its meaning “uv-distance” in both captions.

Figure 3 shows the jet collimation profile based on the measured jet width which includes the new data indicated by red filled circles. Regarding presentation, there are no error bars on the red points, or indication of the generic error associated with these measured values in the figure caption. Is this error within the symbol size? This information should be added to the figure or stated in the figure caption. Also, I presume that “observed distance” labeled on the abscissa means “measured”, or are these deprojected values? Additionally, the caption should include references for the top 3 symbols in the upper left portion of the figure. Further, I would have expected to see a comparison to results in Kim et al. 2018 (cited elsewhere but not in this specific context). Is there some reason why it was not included? Regarding interpretation, the deviations of these points from the simulation are an important new result, but the suggested explanation of a wind to account for a broader-than expected profile in the spatial region inward of 10 Schwarzschild radii is only marginally explored. Is stacking expected to fill in the emission and widen the emission profile as more high-resolution mm data are obtained in future experiments? If so, this will increase the discrepancy between data and simulation identified here and enhance the need for a plausible physical explanation.

Regarding the presentation of the figure, we have added a statement in the figure caption regarding the error bar of the new data. We also changed “observed distance” to “measured distance” and added three references for the top 3 symbols in the upper left portion of the figure. Our results are

consistent with the Kim et al. 2018 results, but with much improved uncertainties. With the improved resolution we can also go closer to the black hole. The reason we did not include this work for a direct comparison is because of the slightly different way of measuring the jet width. In our work, we adopted the distance between the outer edges of the FWHMs of the two outermost Gaussians fitted to the transverse slices at each measured jet axial distance, as adopted in the three cited works. In Kim et al. 2018, the width is measured as the distance between the peaks of the two outermost Gaussians.

Regarding the interpretation, we think the wind is currently the best possible solution, but a more detailed study of the wind and other possibilities to account for the broader-than-expected profile is beyond the scope of this work and therefore should be discussed in a more dedicated theory work. Future experiments with higher sensitivity could further investigate the discrepancy between data and simulation identified here. Stacking of future high-resolution images could help to increase the discrepancy, but may also suffer from some time variability.

Line 193 says that an outcome of the work is to constrain the energy distribution of the electrons responsible for the millimeter emission. The SED shown in the supplementary material (Figure S15) pertains to the electron distribution and would be better placed in the main body of the paper (as Figure 3 before the collimation profile.)

We thank the referee for this suggestion. However, we think moving this figure to the main body would change our emphasis of the paper, so we would prefer to keep this figure where it is.

In Figure S2c some other color scheme should be adopted for this plot. It is difficult to distinguish the blue and the black points, especially where the density of points is relatively high.

Done. We have changed the blue color to red.

Were the data for the calibrator 3C 279 used in the analysis? If so what was their role?

We used 3C 279 for confirming the overall amplitude gain correction discussed in the “Gain correction” section in the supplementary information Section 3 (original supplementary line 229).

In Figure S15 do the authors have any suggestions for why the data labeled “historical” are so different from the more recent data which are very consistent?

We note that these “historical” data are the SED of the core at a scale of ~ 0.4 arcsec, which include the contribution of the large scale jet emission. We have clarified this in the figure caption.

While the paper has been unusually well-prepared, I noted a few places in the text where terms are not defined for the reader, or the idea is so briefly stated that it is obscure. These are listed below.

line 150: Please clarify what is meant by “two railed jet emission”?

We have reworded it as “edge-brightened jet emission”.

line 191-192: what is the “funnel wall jet”.

We have reworded it as “funnel wall jet (jet sheath)”.

lines 222- 225: Higher concentration would affect intensity. How does it affect size?

Thanks for this question. Here we mean a predominance of non-thermal electrons in the jet. In the revised manuscript, this paragraph has been rewritten.

line 444: The comment that the ringlike structure can be explained by opacity effects is obscure and needs further explanation.

We thank the referee for this comment. The original text that the ringlike structure can be explained by opacity effects was a bit misleading. We have clarified it now.

supplement line 131: “Possibilities” is an odd choice of word. Would “configurations” be better?

Thank you for your suggestion. Done.

supplement line 142: “Basis” could be replaced with “mode”.

Done.

The text in supplement lines 213 and 430 both contain the word “opacity”. Presumably the authors

mean opacity in the sky in the first usage, and opacity in the jet outflow in the second. It would be helpful to the reader for the authors to clarify what they have in mind.

This is a very good point. We have revised the text following the suggestions.

Referee #2 (Remarks to the Author):

This paper presents new GMVA + ALMA + GLT observations at 3.5mm of the core of M87 and compares and contrasts the results with the 2017 EHT images at 1.3mm. The work is original, topical, well-presented, and of general interest, and so deserving of consideration for publication in Nature. I append below a list of recommendations, suggestions, and corrections (many of which are relatively minor) which I believe will help improve the paper.

We sincerely thank this referee for carefully reading the manuscript and the positive overall review. We appreciate the very helpful recommendations, suggestions, and corrections.

My main concern is that the conclusions of the paper are, by necessity, based on a comparison of the 2017 EHT imaging with the GMVA+ALMA+GLT observations made one year later, under the assumption that the accretion disk has not changed significantly in the meantime. It is clear that the region around the core does change – the paper notes this is the most sensitive 3.5mm observation made, yet there is no sign of the counter-jet reported at earlier epochs. And we know that new jet components are on occasions ejected from the core. Accretion towards the black hole is unlikely to be steady, and the accretion disk could plausibly swell up and shrink back down from time to time, and the core flux density change as a result. The authors could mount a case, based on multi-wavelength monitoring, that M87 appears to have been in a similar state in 2018 and 2017, or perhaps present simulations that suggest the accretion disk size does not change dramatically under a variety of circumstances. I would need to see some consideration of this point before recommending the paper be accepted for publication.

We thank the referee for this comment and understand his or her concern. Since the 1.3mm ring-like structure imaged by the EHT is believed to be associated with the black hole shadow produced by strong gravitational lensing, it is not expected that its overall size/diameter can change measurably in one year. Indeed, modeling of proto-EHT observations and the 2017 EHT data (Wielgus et al. 2020, ApJ, 901, 67) showed a consistent ring diameter from 2009 to 2017 (although the azimuthal pattern, i.e., the position angle of the intensity peak on the ring may change). GRMHD simulations (EHT collaboration et al. 2019, ApJL, 875, L5) have also shown that the ring-like structure at 1.3mm is dominated by the photon ring and is therefore relatively insensitive to the details of the accretion

flow and jet physics. So the 1.3mm ring diameter in 2018 is expected to be measurably the same as in 2017.

In addition, ALMA observations at 1.3mm by Goddi et al. 2021 (ApJ, 910, L14, Table 3) showed that M87's total flux density decreased from April 2017 to April 2018 by ~10%. Such a small flux change should not lead to a dramatic change of the ring diameter (see Fig. 13 in Wielgus et al. 2020, ApJ, 901, 67). The observed flux density decrease by ~10% shows that the source is not flaring in 2018. Therefore, it is unlikely that the accretion flow swelled up in 2018.

In the revised manuscript, we have added a short discussion that the accretion flow is not expected to have changed significantly between 2017 and 2018.

I note that this paper, like the EHT imaging of M87, presents images that are averaged over an ensemble of plausible images produced independently by "several" members of the team. Perhaps it is not possible to be more precise about exactly how many independent analyses were conducted, but "several" is quite vague. Nine authors are mentioned in the "Author Contributions" as having worked on the image reconstruction and model fitting: perhaps that is sufficient, but I would encourage more quantitative statements where that is possible.

We thank the referee's comment. In the supplementary information section, we have specified the number of independent analyses.

Line 108:

"In 2017, Event Horizon Telescope observations of M87 at the wavelength of 1.3 mm revealed a ring-like structure which was interpreted as gravitational lensed emission around a central black hole."

I think this would read better moving "In 2017" into the sentence, e.g., "Event Horizon Telescope observations of M87 in 2017, at a wavelength of 1.3 mm, revealed a ring-like structure which was interpreted as gravitational lensed emission around a central black hole."

We thank the referee for this suggestion and have reworded this sentence as suggested.

Line 113:

"High resolution imaging reveals a ring-like structure of $8.4_{-1.1}^{+0.5}$ Schwarzschild radii in diameter, ~50% larger than that seen at 1.3 mm. The outer edge at 3.5 mm is larger than that at 1.3 mm."

This would read better as

"High resolution imaging reveals a ring-like structure $8.4_{-1.1}^{+0.5}$ Schwarzschild radii in

diameter, ~50% larger than that seen at 1.3 mm, with the outer edge of the ring at 3.5 mm also being larger than that at 1.3 mm.”

Thank you. We have revised this sentence following the suggestion.

Line 117:

“The new images show that the edge–brightened jet connects to the accretion flow and the black hole.”

I advocate either “...connects to the accretion flow around the black hole.” or “...connects to the accretion flow.” The authors postulate in line 149 that the double structure may be the footpoints of the jet which (line 241) are anchored at the event horizon, but also provide good evidence that the effects of limited (u,v) coverage are important, so I would recommend not claiming the images show the jet connecting to the black hole (assuming that the event horizon can be considered the black hole itself...).

Done. We have reworded it as “...connects to the black hole’s accretion flow”.

Line 124: “GMVA project code ML005”

This information is given in the Supplementary Methods, which is the better place for it, so can be omitted here.

Agreed. We have dropped the text.

Line 126:

“In Fig. 1, we display the resulting maps of M87, revealing a triple-railed jet emerging from a spatially resolved radio core which appears as a faint ring...”

I suggest “triple-ridged” rather than “triple-railed”

Thanks for the suggestion. Done.

Line 136:

“This was done with and without subtracting the outer jet emission (Supplementary Information, Section 3).”

That is quite reasonable, but it would be appropriate to add a few words of explanation as to why it was considered necessary to do this.

Done. We have added a few words of explanation.

Line 138:

“...we arrive at the conclusion that the structure seen with the nominal resolution, indeed is the signature of an underlying ring-like structure with a diameter of 64_{-8}^{+4} μas (see Supplementary Information for details), which becomes best visible in slightly super-resolved images.”

I suggest “is most apparent” rather than “becomes best visible”

Done.

Line 143:

“ $R_s=2G-M/c^2$ ”

Remove the dash between G and M

Thanks, this typo has been fixed.

Line 149:

“On the other hand, this double structure may also mark the two 'footpoints' of the northern and southern ridge of the two railed jet emission, which is seen further downstream.”

It is previously noted that the jet is triple-ridged, so some consideration of why the central ridge does not have an associated footpoint may be appropriate, within this paragraph or the paragraph beginning on Line 163. It would be useful to point out here that the EHT imaging shows no jet emission at all, due to the limited region they considered in their imaging. It should also be noted whether (and why) the EHT image in Figure 2d was reconstructed adopting the same constraints as the EHT team (and stating what they are).

We thank the referee's suggestion. For a poynting-flux-dominated jet spine we do not expect to see the central ridge to be connected all the way down to the ring-like structure. However, the central ridge is very faint and cannot be traced very well in our image. It is only suggestive that the back extrapolation of the central ridge points towards the black hole position. We would defer the investigation of whether or not the central ridge is connected to the ring-like structure and their implications to a future study.

We noted that the published 1.3mm images did not reveal the inner jet emission, but some recent reanalysis by other groups found some possible extended emission outside of the 1mm ring. Therefore, we added a note on this point.

In Figure 2d, the image was reconstructed using the same data and pipeline as published in the EHT paper. We have clarified it in the figure caption.

Line 156:

“This larger size at 3.5 mm is not caused by observational effects...”

It would be appropriate to state, or give examples of, the potential “observational effects” considered here.

Thanks for this suggestion. We have added examples of the potential observational effects including calibration and (u,v)-coverage.

Lines 157 and 160:

“...uv-distance...”

Would be better as (u,v)-distance

Done.

Line 169:

“The intensity of the central ridge is ~60% of that of the outer jet ridges, suggesting the presence of a central spine, which points towards the ring centre.”

At face value, the sentence seems to imply it is the lower intensity of the central ridge that suggests it is a central spine – if that is indeed the intention further explanation should be given. The location of the central ridge is sufficient to assume it is the jet spine, in which case “The location of the central ridge, which has an intensity ~60% of that of the outer jet ridges, suggests the presence of a central spine...” would suffice.

But does the central spine “point towards the ring centre” or away from it? Perhaps “which emerges from the ring centre” or “which is aligned with the ring centre” would be better, although in fact extrapolation of the innermost portion of the central ridge does not appear to pass through the ring centre. While “spine and sheath” models for jets have been considered for some years, to the best of my knowledge M87 is the only source which appears to display both edge-brightened sheath and interior spine. (And this result is arguably more robust, as it does not entail comparison with the EHT observation a year earlier.) The GMVA+ALMA+GLT image is much more convincing than those in the cited refs 5 & 7. It is probably beyond the scope of this paper to consider in more detail, but I trust a paper exploring the implications of this jet structure is planned.

Thanks for this great suggestion. We agree that it is not the lower intensity, but the location of the central ridge that suggests it is a central spine. We have therefore revised it as “The location of the central ridge, which has an intensity ~60% of that of the outer jet ridges, suggests the presence of a central spine, which emerges from the ring centre.” Indeed, exploring the implications of the triple-

ridged jet structure is an interesting topic, but it is beyond the scope of this particular paper and shall be discussed elsewhere.

Line 175:

“In order to understand the physical nature of the 3.5 mm ring-like structure, we first need to estimate the position of the black hole. It is well-known that the location of the unresolved core in VLBI images of AGN jets shows a frequency dependence (the so-called "core shift", e.g., ref.10). For a synchrotron self-absorbed homogeneous jet the VLBI-core shifts upstream when observing at higher frequencies. Previous core shift measurements of M8711 also showed this dependence. From these measurements, the position of the jet apex was extrapolated to be located at a projected distance of $41 \pm 12 \mu\text{as}$ or $5.4 \pm 1.6 R_s$ (adopting the black hole mass and distance from EHT results⁴) upstream of the 7 mm (43GHz) radio core. Extrapolating the frequency dependent core position further would place the black hole at a projected distance of $\lesssim 3 R_s$ ($\sim 21 \mu\text{as}$) upstream of the VLBI core at 3.5 mm (where we neglect a small position offset between jet apex and the BH position). It is therefore natural to assume that the centre of the ring-like structure which we have observed at 3.5 mm is close to the actual black hole position.”

I think all bar the last sentence of this paragraph could probably be omitted. The “core shift” effect is relevant for the classic morphology described in Lines 163—166 but, as noted, the 3.5mm image considered here does not have an unresolved core. As we see a ring, it is natural to assume the black hole lies at its centre. The rest of the paragraph could be moved to the Supplementary Methods, or possibly omitted entirely.

We agree with the referee’s suggestion. We have merged the last sentence with the next paragraph and have dropped the rest of this paragraph.

Line 187:

“Given the typical brightness temperature of $\sim 10^{10}$ K, synchrotron emission is believed to be responsible for the 3.5 mm ring-like structure.”

As the measured brightness temperature is stated on line 161, this sentence would be better reworded as something like

“Given the measured brightness temperature of $\sim 10^{10}$ K is typical for that of AGN cores, synchrotron emission is believed to be responsible for the 3.5 mm ring-like structure.”

Done.

Line 194:

“Recent EHT observations have confirmed the nature of the accreting black hole in M87 to be in the low-Eddington regime, ...”

I suggest “The 2017 EHT observations...” rather than “Recent EHT observations...”

Agreed.

Line 200”

“The boundary between the accretion flow and jet is defined as the surface where the magnetic energy density equals to the fluid's rest-mass energy density...”

Either “...density equals the...” or “...density is equal to the ...”

Thanks for the suggestion. This has been fixed.

Line 221:

“From the above discussion, the thermally-distributed electron model explains the larger ring diameter at 3.5 mm, under the constraints of the 1.3 mm and 3.5 mm flux density.”

At a minimum this should be “... 1.3 mm and 3.5 mm flux densities”, but better would be

“... under the constraints of the (2017 April) 1.3 mm and (2018 April) 3.5 mm flux densities.”

This would lead naturally to some consideration of whether, and how much, the core of M87 may have evolved in the intervening period, and the effect of such variability on the conclusions drawn.

We agree with the referee. This has been merged into the new paragraph in response to the major comment above.

Line 246:

“We point out that a laterally expanding wide opening angle BZ jet launched from a MAD accretion flow...”

Need to spell out MAD at its first usage.

Agreed. Here we have rephrased it to “...from a strongly magnetised accretion flow (i.e., the so-called magnetically arrested disk, MAD)”

Line 319:

“Data availability

The ALMA raw visibility data can be retrieved from the ALMA data portal under the project code 2017.1.00842.V. The reduced data that support the results of this study are available from the corresponding authors upon reasonable request.”

Some additional detail would be useful – is it the ALMA phased array data that is available? Are the ALMA internal baselines also available? Perhaps “The reduced data...” could be expanded to “The reduced GMVA + ALMA + GLT data...”, assuming that is correct.

We thank the referee's comment. Since the disk modules were normally recycled for new observations after some time, the ALMA phased array data are not available. However, the ALMA interferometric data (i.e., the internal baseline data) are available. We have clarified the availability of the VLBI data and revised the data availability statement as "The ALMA internal baseline data can be retrieved from the ALMA data portal under the project code 2017.1.00842.V. The reduced GMVA + ALMA + GLT data that support the results of this study are available from the corresponding authors upon reasonable request."

Figure 1:

Line 422: Expand "p.a."

Line 431: "For each panel, the colour map denotes the brightness temperature T in Kelvin, which is related to the flux density S in Jy ..."

When spelled out in full, a physical unit named after a person is not capitalised (even though its abbreviated form is!). So it should be "... brightness temperature T in kelvin ..." (and similarly could be "... the flux density S in jansky")

It would be appropriate to note in the caption that these images are the averages of the best fitting images produced independently by several team members, referring to the Supplementary Methods for more details.

We have expanded "p.a.". We thank the referee for pointing out the capitalisation issue on the physical unit, which has been fixed. Following the referee's suggestion, We have added a note in the caption on the final image being the result of image averaging.

Figure 3:

The inset or legend to the figure would be better with "ref 16" or "Asada and Nakamura 2016" rather than "AN16", and similarly for H13 and H16.

Line 458: "The light-grey shaded area denotes the outermost streamlines of the parabolic jet's envelope from theoretical simulations ... that are anchored at the event horizon for a range of black hole spins (dimensionless spin parameters: $a = 0.0 - 0.9$)."

It would be of interest to note which extreme of the shaded grey area corresponds to which extreme of the spin parameter.

The legend to this figure has been revised as suggested. We have added one sentence to explain which extreme of this shaded grey area corresponds to which extreme of the spin parameter.

Supplementary Methods

=====

Line 109:

"M87 was observed by the Global Millimetre VLBI Array (GMVA) in concert with the phased Atacama

Large Millimetre/submillimetre Array (ALMA) and the Greenland Telescope (GLT)¹ at 3.5 mm (86 GHz; GMVA project code ML005). The GMVA is composed of the eight Very Long Baseline Array (VLBA) antennas equipped with 3.5 mm receivers, ...”

It would appropriate to add how many of ALMA’s dishes were phased up for these observations (only 37 of the 54 12-m dishes were phased up for the 2017 EHT observations), and also which array configuration ALMA was in at the time. It would also be useful to explicitly state which two VLBA antennas do not have 3.5mm receivers.

Agreed. We have included the number of phased dishes and array configuration in the text. The two VLBA stations (i.e., Saint Croix and Hancock) that do not have 3mm receivers have been mentioned.

Line 153:

“We removed the instrumental polarisation effects from the GLT by applying, in the frame of the Radio Interferometer Measurement Equation...”

“...in the frame-work...” would be better

Done.

Line 190:

“We then derived and applied bandpass corrections using calibrator scans and performed the a priori amplitude calibration in the standard manner, using measured system temperatures and gain curves and including opacity corrections for each station.”

Consider adding a brief description of what measurements the opacity corrections are based on.

Done. We have added a brief description for opacity correction.

Line 203:

“and closure phases of $\sim 180^\circ$ on ALMA-US-EU and ALMA-US-GLT triangles.”

The use of US and EU is probably clear, but perhaps spell out in full for clarity.

We have spelled them out to “ALMA-USA-Europe and ALMA-USA-GLT”.

Line 225:

“This was done independently by several authors.”

This is an important point. Is there any reason the number of independent authors (or teams) could not be given?

Done. We have given the number of authors/teams.

Line 238:

“are in good agreement with near in time VLBI observations of the Korea VLBI Network (KVN) at 3.5 mm...”

should be “Korean VLBI Network”

Done.

Line 276:

“We utilised weighted-L1 (wL1), total squared variation (TSV), and total variation (TV) regularizers for sparse imaging (see ref.9 for their mathematical definitions).”

It would be useful to include here a little more detail, for example

“...total squared variation (TSV, which favours smooth edges)...”

This is a good point. Done.

Line 298:

“For the other hyper parameters, we tried multiple values.”

A little more detail on how these multiple values were chosen would be useful.

Done. We have provided the used values.

Line 301:

“Since the chi-square values of all images now were reasonably low, a chi-square cutoff was not applied.”

should be “chi-squared”

Fixed.

Line 303:

“The resulting optimal set of images (216 images in total)...”

It would be of interest to note how many images in total were generated (and therefore what fraction were, and were not, considered optimal)

We have added the total number of generated images.

Line 309:

“We mainly used this source because the GLT did not observe 3C279.”

Presumably as 3C279 is too far south to observe with the GLT – but if that’s the case, then explicitly saying so would be better.

Agreed. We have revised it as “We mainly used this source because the GLT did not observe 3C279 due to its low declination.”

Line 324:

“With our high-resolution image, we also found that there is emission upstream of the brightness peak. Full details of these results will be published elsewhere.”

This is interesting, but with the saturated colour scale in Figure S5b it is not readily apparent. In any case it is peripheral to the main focus of this paper, so these two sentences could possibly be dropped.

We agree with the referee. We have dropped these two sentences.

Line 346:

“We postulate that this difference originates from: (i) the limited (u,v)-coverage of our data, and (ii) that the RML technique is better suited for super-resolution imaging than CLEAN.”

This would read better as “...(ii) the fact that the RML...”

Thank you. Done.

Line 354:

“One is based on the real (u,v)-coverage, 10-second averaged data and the other is generated by adding artificial baselines based on two new stations (VLBA MK and KVN Yonsei).”

VLBA MK is not really new – it participated in the observation but no fringes were found for M87.

Perhaps “two additional stations”? Or “adding artificial baselines to VLBA MK and KVN Yonsei.”

Thank you for the suggestion. We have reworded it as suggested.

Line 381:

“Using this tentative position as the origin, we unwrapped the image azimuthally and obtained radial profiles for each azimuthal angle between 0 and 360°.”

Does this mean 360 radial profiles in one-degree steps?

Thanks for this question. We sampled 100 radial profiles in 3.6-degree steps between 0 and 360°. We have revised the text accordingly.

Line 389:

“There are some radial profiles where the brightness monotonically decreases with radius (see, e.g., Fig. S6a).”

This is not clear to me – I do not see any radial profiles that decrease monotonically with radius from the centre of the ring in Fig. S6a!

Good catch, Fig. S6a is indeed not the correct example. We have revised it to Fig. S6c.

Line 412:

“This fitting was applied to both the full self-calibrated dataset and to the jet-subtracted dataset. In each case, the data were averaged to 420 seconds.”

It would be appropriate to add a justification for this choice of averaging interval.

Done. The data were average to scan length (420 seconds).

Line 451:

“In addition, we note the formal size of the inner depression as described by r_{in} is about 1.7 times larger than the inner radius of $\sim 13 \mu\text{as}$ measured from the EHT data²¹.”

I suggest “best-fit size” (or “formal best-fit size”)

Done.

Line 454

Altogether, the analysis presented above suggests the presence of a ring-like structure in the centre of M87 which has a mean radius of $\sim 67 \mu\text{as}$ and a width of $\sim 22 \mu\text{as}$.

“radius” should be “diameter”

Thanks for catching this mistake. We have corrected it.

Line 486:

“We performed an axisymmetric 2D GRMHD numerical simulation for magnetised plasma around a rotating black hole (dimensionless black hole spin parameter = 0.9) by using the public code HARM”

Please add a justification for this choice of spin parameter

Done. We have added a justification for the choice of the spin parameter.

Line 499:

“For modelling the core emission and image structure of M87, we consider a black hole of 6.5 billion solar masses ...”

I recommend scientific notation, 6.5×10^9

Done.

Figure S7:

“Simulated (u,v)-coverage for the 2018 observations of M87 (blue) and for the artificially added baselines based on VLBA MK in Hawaii, USA and KVN Yonsei in South Korea (red).”

Perhaps

“Simulated (u,v)-coverage for the 2018 observations of M87 (blue) and for the additional baselines to VLBA MK in Hawaii, USA and KVN Yonsei in South Korea (red), added for the purpose of investigating the effects of (u,v) coverage on the results.”

Done.

Figures S9 and S14.

The insets for both figures should use “With jet” and “Without jet” rather than “W jet” and “W/O jet”

Done. Both figures have been updated accordingly.

References:

Reference 6: The volume number (which should appear in bold) is 285, and the page number (which should not be in bold) is 109.

Reference 14: The article id. of this paper is 95, not 5

There appears to be some inconsistency in both reference lists: in some cases both the first and last page numbers of articles are given, in other cases only the first page number is given (and of course in some cases only the article id. is required)

Thank you, very good catch. We have fixed references 6 and 14 as well as the inconsistency in both reference lists. Both the first and last page numbers of articles are now given (unless the first ‘page’ is an article number).

Reviewer Reports on the First Revision:

Referees' comments:

Referee #1 (Remarks to the Author):

This is a review of the revised version (V2) of the manuscript and supplementary material produced by the authors in response to the comments by myself and a second reviewer. The revised version includes both modifications of the original text and textual additions included in order to clarify some points raised by the review comments. As the authors themselves state, the most remarkable feature of the new images and the paper itself is the spatially-resolved core feature showing the ring-jet connection in the new data which they have carefully analyzed. I noted in my earlier review that the paper is worthy of publication in Nature, and I believe that the authors have for the most part responded to the questions and concerns stated in my earlier review. I include below a few additional comments and questions for their consideration.

One of the interesting results from this work is the evidence presented for the presence of a wind based on the shape of the jet profile and cited in lines 122-124 of the abstract. As I noted earlier, the necessity for a wind to account for the observed jet width down the jet is an important result with implications for jet formation and evolution which could receive more attention within the paper text. While the authors chose not to expand this section in their response, they might consider adding one or two more recent references e.g. Blandford and Globus, 2022 Galaxies, vol.10, 89 to bring this section more up to date.

In lines 138-139 “full assessment of the parameters of the core structure” seems to be an overambitious statement. The authors have examined several scenarios and selected the most likely one.

I find the wording of line 171 regarding the triple ridge jet structure confusing as stated. With such a

difference in physical scale between the physical regions probed by these data sets, I am a bit unclear on how they connect this triple structural feature across these scales. The suggestion that they are physically related appears implied to me by the text provided.

Do the authors have any plausible explanation which could be added regarding the lack of a measurable counter-jet feature (lines 176-177)? This structure would have been expected in the new imaging based on the previous observations.

In the text in lines 213-217 I find the logical flow of the argument presented difficult to follow. The authors state that the ring size is constant at 1.3 mm over the one year period. They then state that the ring is larger at 3.5 mm than at 1.3 mm. How does it follow from these two pieces of information that the proposed accretion flow has high opacity as they state here and in line 120 of the abstract? Isn't this conclusion in fact based on a comparison with their model predictions (line 205)? Also, can they please clarify what "high" means here.

In Figure 3 do the authors have an explanation for the origin of the well-defined discontinuity/dip apparent in the plot where the grey and red points join and relative to the trend exhibited by the combined data shown (jet width as a function of jet distance downstream).

In this figure it is difficult for the reader to distinguish the grey points denoting the Hada 2013 data from the Hada 2016 data in the region where the points overlap. A more distinct choice of symbol color/shape would be helpful to the reader.

Finally, I noted several minor grammatical errors which I presume will be corrected by the Nature editors or by the authors themselves before publication. I list some of these below as examples.

line 133 spatially-resolved [missing hyphen]

line 134 in THE north- [missing the word THE]

line 188 well-described

line 190-191 by the jet or BY the accretion flow [missing the word BY to complete the parallel construction]

line 195 1; [need a semicolon]

line 222 black hole-driven [hyphen needed]

line 307 in THE standard ...

line 320 comma needed after the word "members"

line 325 publicly-available [need a hyphen]

line 364 colormaps ... ARE available...

line 393 acknowledges THE program ...

Referee #2 (Remarks to the Author):

The authors are thanked for carefully addressing the reviewers' comments, which has resulted in an improved paper that I recommend be accepted for publication.

The only minor correction required is in the caption to Figure S4, which refers to magenta and black symbols, whereas magenta and blue symbols are used.

Author Rebuttals to First Revision:

Referees' comments:

Referee #1 (Remarks to the Author):

This is a review of the revised version (V2) of the manuscript and supplementary material produced by the authors in response to the comments by myself and a second reviewer. The revised version includes both modifications of the original text and textual additions included in order to clarify some points raised by the review comments. As the authors themselves state, the most remarkable feature of the new images and the paper itself is the spatially-resolved core feature showing the ring-jet connection in the new data which they have carefully analyzed. I noted in my earlier review that the paper is worthy of publication in Nature, and I believe that the authors have for the most part responded to the questions and concerns stated in my earlier review. I include below a few additional comments and questions for their consideration.

We thank the referee for the positive review of the revised version of the manuscript and the additional comments.

One of the interesting results from this work is the evidence presented for the presence of a wind based on the shape of the jet profile and cited in lines 122-124 of the abstract. As I noted earlier, the necessity for a wind to account for the observed jet width down the jet is an important result with implications for jet formation and evolution which could receive more attention within the paper text. While the authors chose not to expand this section in their response, they might consider adding one or two more recent references e.g. Blandford and Globus, 2022 Galaxies, vol.10, 89 to

bring this section more up to date.

We thank the referee for this suggestion. We have added the suggested reference.

In lines 138-139 “full assessment of the parameters of the core structure” seems to be an overambitious statement. The authors have examined several scenarios and selected the most likely one.

We agree with the referee. We have reworded it.

I find the wording of line 171 regarding the triple ridge jet structure confusing as stated. With such a difference in physical scale between the physical regions probed by these data sets, I am a bit unclear on how they connect this triple structural feature across these scales. The suggestion that they are physically related appears implied to me by the text provided.

We understand the referee’s concern. We have rephrased the text to remove the implication that the triple structural features on those scales are physically related.

Do the authors have any plausible explanation which could be added regarding the lack of a measurable counter-jet feature (lines 176-177)? This structure would have been expected in the new imaging based on the previous observations.

Done. We have added some plausible explanations for the lack of the measurable counter-jet emission.

In the text in lines 213-217 I find the logical flow of the argument presented difficult to follow. The authors state that the ring size is constant at 1.3 mm over the one year period. They then state that the ring is larger at 3.5 mm than at 1.3 mm. How does it follow from these two pieces of information that the proposed accretion flow has high opacity as they state here and in line 120 of the abstract? Isn’t this conclusion in fact based on a comparison with their model predictions (line 205)? Also, can they please clarify what “high” means here.

We agree with the referee and have rephrased this part to make the logical flow clear. We also clarified what “high” opacity means.

In Figure 3 do the authors have an explanation for the origin of the well-defined discontinuity/dip

apparent in the plot where the grey and red points join and relative to the trend exhibited by the combined data shown (jet width as a function of jet distance downstream).

The discontinuity/dip in the region where the grey and red points join may be caused by local/temporal features/patterns in the jet. We note that the power law indices for the jet width profile (0.56-0.58) found using individual dataset in the cited works are fully consistent with the trend seen in the combined data (with a power law index of 0.58, as indicated by the dashed line in Figure 3). Given the relatively large uncertainties of the corresponding points and the non-simultaneity of these observations, we think discussion the origin of these features is beyond the scope of this paper. Future simultaneous observations at multiple frequencies should allow a more robust investigation of such features.

In this figure it is difficult for the reader to distinguish the grey points denoting the Hada 2013 data from the Hada 2016 data in the region where the points overlap. A more distinct choice of symbol color/shape would be helpful to the reader.

Fixed.

Finally, I noted several minor grammatical errors which I presume will be corrected by the Nature editors or by the authors themselves before publication. I list some of these below as examples.

line 133 spatially-resolved [missing hyphen]

line 134 in THE north- [missing the word THE]

line 188 well-described

line 190-191 by the jet or BY the accretion flow [missing the word BY to complete the parallel construction]

line 195 1; [need a semicolon]

line 222 black hole-driven [hyphen needed]

line 307 in THE standard ...

line 320 comma needed after the word "members"

line 325 publicly-available [need a hyphen]

line 364 colormaps ... ARE available...

line 393 acknowledges THE program ...

We thank the referee for pointing out these grammatical errors, which have been fixed.

Referee #2 (Remarks to the Author):

The authors are thanked for carefully addressing the reviewers' comments, which has resulted in an improved paper that I recommend be accepted for publication.

The only minor correction required is in the caption to Figure S4, which refers to magenta and black symbols, whereas magenta and blue symbols are used.

Done.